# Probiotic Bacteria Cannot Mitigate the Adverse Effects of Radioactive Iodine-131 Treatment

**DOI:** 10.3390/cancers15030740

**Published:** 2023-01-25

**Authors:** Seyed Mohammad Javad Mortazavi, Saba Nowroozi, Masoud Haghani, Zinat Zarrini-Monfared, Farshid Gheisari, Lembit Sihver

**Affiliations:** 1Medical Physics and Engineering Department, School of Medicine, Shiraz University of Medical Sciences, Shiraz 71348-14336, Iran; 2School of Medicine, Shiraz University of Medical Sciences, Shiraz 71348-14336, Iran; 3Radiology and Radiobiology Department, School of Paramedical Sciences, Shiraz University of Medical Sciences, Shiraz 71348-14336, Iran; 4Department of Medical Physics & Biomedical Engineering, School of Medicine, Tehran University of Medical Sciences, Tehran 19835-178, Iran; 5Nuclear Medicine Department, School of Medicine, Shiraz University of Medical Sciences, Shiraz 71348-14336, Iran; 6Department of Radiation Dosimetry, Nuclear Physics Institute of the CAS, 180 00 Prague, Czech Republic; 7Technische Universität Wien, Atominstitut, 1020 Vienna, Austria

**Keywords:** nuclear medicine, radioiodine therapy, probiotic bacteria, iodine-131

## Abstract

**Simple Summary:**

Thyroid carcinoma is the most common cancer of the endocrine system and accounts for 12% of all cancer cases in adolescents in the United States. In this paper, we present a double-blind, randomized, placebo-controlled clinical trial aimed at evaluating the effect of probiotics supplementation in reducing the acute side-effects of radioiodine therapy in DTC patients. The probiotics’ effectiveness was confirmed for dry mouth and taste loss or change when it was administered prior to the radioiodine treatment. The benefit was not confirmed for other radiation-induced complications such as pain and swelling in the neck, nausea and vomiting, salivary gland swelling, and diarrhea. Further large-scale clinical trials are warranted to improve our knowledge of this quickly evolving field.

**Abstract:**

Thyroid carcinoma is the most common cancer of the endocrine system, accounting for 12% of all cancer cases in adolescents in the United States. Radioiodine therapy plays a key role in differentiated thyroid cancer (DTC) treatment. This double-blind, randomized, placebo-controlled clinical trial was aimed at evaluating the effect of probiotics supplementation in reducing the acute side-effects of radioiodine therapy in PTC patients. Fifty-six patients were randomly divided into four groups: one placebo and three intervention groups. The probiotics product used in this study was LactoCare (ZistTakhmir Co., Tehran, Iran), a multi-strain commercially available symbiotic containing 12 strains of probiotic species including Lactobacillus strains, Bifidobacteria strains, and Streptococcus thermophilus, plus Fructo-oligosaccharides as the prebiotic. Group 0 was our placebo group (no probiotics), while the other three groups received probiotics capsules for 2/4 days, starting only 2 days prior to radioiodine therapy, only 4 days after radioiodine therapy or 2 days prior and 4 days after radioiodine therapy. Six patients were withdrawn during the study because of poor compliance or at their own request. The symptoms reported by patients including data about the incidence and duration of each complication were recorded. The probiotics’ effectiveness was confirmed for dry mouth and taste loss or change when it was administered prior to the radioiodine treatment. The benefit was not confirmed for other radiation-induced complications such as pain and swelling in the neck, nausea and vomiting, salivary gland swelling, and diarrhea. Further large-scale clinical trials are warranted to improve our knowledge in this quickly evolving field.

## 1. Introduction

Thyroid carcinoma is the most common cancer of the endocrine system and accounts for 12% of all cancer cases in adolescents in the United States [1]. It is among the top 10 cancer types diagnosed in females, with a lifetime prevalence of 1 in every 55 women [1]. About 90% of the thyroid cancer types are differentiated thyroid cancer (DTC) [2]. DTC is a rare malignancy with excellent outcome and the therapeutic approach includes surgery that is normally followed by radioiodine treatment for the remnant ablation or adjuvant treatment [2]. Radioactive iodine, I-131, was first used for thyroid cancer patients in 1946. and for several decades has played the key role in the treatment of DTC [3]. Despite the recent changes in DTC management, radioiodine therapy is still the therapy of choice for selected patients [4,5,6,7]. The radioactive iodine is administrated (in the form of sodium iodide or potassium iodide) to irradiate and ablate the suspected cancerous tissue that have not been completely resected during the surgery [2]. The treatment decreases the recurrence rate of the disease and improves overall survival. Despite the benefits of the radioiodine treatment, some potential short-term and long-term side effects have been reported. Pain and swelling in the neck, nausea and vomiting, dry mouth, diarrhea, salivary gland swelling and pain, headache and fatigue complications are among the most common adverse effects [8,9].

Gastrointestinal (GI) toxicities associated with radiation exposure are caused by oxidative stress, during which reactive oxygen species are produced in excessive amount to be naturally moderated. Oxidative stress can cause GI damage through numerous pathways [10]. Ionizing radiation can damage microorganisms that inhabit the GI system, reducing their abundance and diversity, and in turn, causing an increase in the pathogenic species population [11]. In addition, the damage to the epithelium of the GI system leads to mucositis and enteritis with various symptoms, reducing the quality of life and, in severe cases, interrupting the optimal cancer treatment and extending the treatment time.

Different management approaches have been suggested for radiation-induced gastrointestinal toxicities [12,13]. The use of food supplements such as prebiotics and probiotics have shown promising results in various studies [14,15]. Probiotics can have prophylactic and therapeutic benefits for acute radiation side effects through several radioprotection mechanisms. These mechanisms include the manipulation of the damaged microbiota of the GI tract, their antioxidant capacities, and the ability to influence the radiation-induced immune response [14]. 

Various clinical trials have evaluated the effectiveness of probiotics in reducing the radiation induced GI toxicities, with supportive results [16,17,18,19]. There are also several recent systematic reviews and meta-analyses of RCTs showing the benefits of probiotics and synbiotics in reducing the incidence of radiation-induced oral and gut toxicities [15,20,21,22]. Despite various supportive evidence, the results are still inconsistent [23]. The reason may be due to the numerous involved factors and the wide range of medical conditions [24]; the probiotics strains and the administered dosage and regimes, radiation-related factors such as dose, technique and the exposed site, and patient-related factors and comorbidities. New human clinical trials may help to improve the evidence in this quickly evolving field.

The aim of this trial is to evaluate the effect of probiotics supplementation in reducing acute side-effects of radioiodine therapy in DTC patients. The focus is on the gastrointestinal tract toxicities, including pain and swelling in the neck, nausea and vomiting, salivary gland swelling, taste loss or change, dry mouth, and diarrhea. The effect of probiotics administration prior to the radioiodine therapy or after the treatment has been evaluated.

## 2. Materials and Methods

### 2.1. Ethics Consideration

This study was approved by the Institutional Review Board of the Shiraz University of Medical Sciences (IRB number: IR.SUMS.MED.REC.1400.621). This clinical trial was registered at the Iranian Registry of Clinical Trials (IRCT) with the identifier IRCT ID of IRCT20220226054126N1. This study was a randomized, double-blind, placebo-controlled clinical trial. Written informed consent (in Persian) was obtained from all patients after explaining the rationale of the project. 

### 2.2. Study Population

DTC patients 18 to 90 years old who were to receive radioiodine treatment following thyroidectomy at the nuclear medicine department of Namazi hospital, Shiraz, Iran from April 2022 to May 2022 with an interest in participating in the project were included. Exclusion criteria were having no interest in participating, use of opioid pain relievers or antibiotics from one week prior to the enrollment and during the treatment course, usage of anti-diarrheal medication, and experiencing acute or chronic side effects prior to the enrollment.

### 2.3. Study Design

A total sample size of 56 patients were enrolled in the trial, from which six cases were withdrawn because of poor compliance or at their request. Patients were informed that participation was voluntary and they were free to refuse to continue at any time during the study. 

The patients were randomly divided into four groups: one placebo and three intervention groups (Figure 1). A random code was allocated to each patient using Excel’s random number generator. All physicians and participants were blinded to the allocation and it was not disclosed during the trial period. The placebo was completely similar in shape, weight, and appearance to the probiotics.

The probiotics product (LactoCare^®^ capsules, ZistTakhmir, Tehran, Iran) is a multi-strain commercially available synbiotic. LactoCare^®^ capsules contain 12 strains of probiotic species including Lactobacillus strains, Bifidobacteria strains, and Streptococcus thermophilus, plus Fructo-oligosaccharides as the prebiotic (with 1 × 10^9^ CFU/capsules).

The groups characteristics were as follows: 

group 0: placebo; 10 patients; routine radioiodine therapy with no probiotics.

group 1: intervention; 13 patients; routine radioiodine therapy, receiving probiotics capsules for 2 days, starting 2 days prior to radioiodine therapy, one capsule once a day after lunch.

group 2: intervention; 14 patients; routine radioiodine therapy, receiving probiotics capsules for 4 days starting from the first day of radioiodine therapy, one capsule once a day after lunch. 

group 3: intervention; 13 patients; routine radioiodine therapy, taking probiotics for 6 days starting 2 days prior to radioiodine therapy up to 4 days after, one capsules once a day after lunch.

The following clinical outcomes were assessed: pain and swelling in the neck, nausea and vomiting, salivary gland swelling, taste loss or change, dry mouth, and diarrhea. The patient-reported symptom assessment with regard to the incidence and duration of the complications were recorded using the scores on a Likert scale questionnaire and at the following time points: 24 h, 72 h, and 120 h after the radioiodine therapy. 

Thyroid-Stimulating Hormone (TSH) levels were measured prior to RAI administration for all patients. The activity of I-131 was monitored by experts until discharge to have an estimate of the biological half-life differences among patients.

## 3. Statistical Analyses

Continuous variables were reported as mean (standard deviation). Categorical variables were reported as number (percentage). A Kruskal-Wallis H test was used to compare the distribution of continuous variables across groups. For categorical variables, the Fisher’s exact test was used. The intervention effects in each group in comparison with the placebo were estimated using a generalized estimating equation (GEE) model. All of the analyses were performed on SPSS version 26 with a two-tailed significance level of 0.05.

## 4. Results

A total of 50 DTC patients over 18 years of age were included in the study and its analysis. The patients were referred to the nuclear medicine department of Namazi hospital, Shiraz, Iran from April 2022 to May 2022 for radioiodine treatment following thyroidectomy.

The baseline characteristics of the population are listed in Table 1. The difference in distribution of these characteristics was not statistically significant from the placebo for all of the intervention groups. 

Table 2 and Table 3 show the administered I-131 activities (mCi) and the number of treatments, respectively. It should be noted that different groups were matched as closely together as possible.

A GEE model was used to evaluate the effectiveness of the probiotic’s capsules on the symptoms at three time points: 24, 72, and 120 h after radioiodine treatment. Table 4 shows the analysis results for intervention groups in comparison with the placebo (group 0). 

The incidence of the side effects in the groups as an absolute number is shown in Table 5 to help better understand the results.

## 5. Discussion

A randomized, double-blind, placebo-controlled clinical trial was performed to assess the effect of probiotics administration on the radiation-induced GI toxicities for DTC patients undergoing radioiodine treatment following thyroidectomy. The clinical outcomes were pain and swelling in the neck, nausea and vomiting, salivary gland swelling, taste loss or change, dry mouth, and diarrhea. The interventions involved probiotics administration before and after radioiodine treatment to observe the specific effects based on the patient-reported symptoms. The results demonstrated no significant benefit in reducing the incidence or change of the symptoms for the intervention groups.

The effect of probiotics in reducing the radiation-induced diarrhea has been investigated with some inconsistency in the results. Several recent systematic reviews and meta-analyses have confirmed the role of probiotics in reducing the radiation-induced diarrhea for external beam RT and brachytherapy of the gynecological and abdominal malignancies [15,21,22,25]. However, the results for other body regions are not as solid [23]. Despite the small *p* value obtained for group 3 (0.078), this study does not confirm the effectiveness of probiotics in reducing diarrhea following radioiodine therapy for DTC patients. The reason may be related to the difference in radiation dose received by the intestines, which causes the damage to be different in severity and outcome; however, other factors such as the small sample size may also be involved. It is worth noting that before RAI therapy, patients usually experience hypothyroidism, which could result in dyspepsia (chest pain), low stomach acid, poor absorption, constipation, gallstones, anemia, and bacterial overgrowth in the small intestine because of the GI hypomotility. However, after RAI therapy, due to radiation-induced enteritis, patients experience diarrhea, nausea, vomiting, and stomach cramps. Given this consideration, after RAI, diarrhea can be linked to the administration of I-131. As Table 4 shows, there were some changes in frequency of the diarrhea, but with a large confidence interval, which makes drawing a firm conclusion difficult. The fact is that diarrhea was not mentioned as a complication by most patients, as shown in Table 5, and this may be a reason for such a large confidence interval in Table 4.

The effectiveness of probiotics in reducing the incidence of dry mouth and oral mucositis has been observed previously [17,26]. Some oral bacteria may be responsible for the occurrence of dry mouth, and probiotics administration may reduce their abundance [17]. In this study, no difference was observed for different intervention groups. However, a fairly small *p* value for group 2 (0.061) might indicate some effectiveness.

There is limited supportive evidence of the benefits of probiotics for reducing nausea and vomiting, though not sufficient to make any firm conclusions [23]. This study did not confirm the effectiveness of probiotics on these complications.

Despite promising results, stating firm conclusions about the effectiveness of probiotics in different GI toxicities is difficult. The heterogeneity in the findings may be related to the many different involved factors. Variabilities in the probiotic composition and dosage, the administered duration, as well as radiation-related factors such as dose, site of irradiation, irradiation technique, and patient-related factors may all cause different outcomes in different trials. 

Probiotics are usually considered to be clinically safe; however, rare adverse effects have been reported in certain cases and for specific strains [27,28]. New preclinical and clinical trials may be considered very helpful in determining the efficiency of probiotic products. However, any claim in support of their benefits should be considered specific to conditions of the specific trials [24].

The limitations of this study include the limited sample size, short duration of the probiotics administration, and short follow-up times. The probiotics dosage and strains need to be optimized. More symptoms and evaluation methods (such as laboratory tests) are suggested. 

More efficient radioiodine therapy using new dosimetry tools to reduce the side effects [29], as well as establishing optimum regulations for the probiotics’ products usage in different complications, may promise a new future for DTC patients to benefit from the advantages of radioiodine treatment.

## 6. Conclusions

The probiotics’ effectiveness was confirmed for dry mouth and taste loss and change when it is administered prior to the radioiodine treatment. The benefit was not confirmed for other radiation-induced complications such as pain and swelling in the neck, nausea and vomiting, salivary gland swelling, and diarrhea. Further large-scale clinical trials are warranted to improve our knowledge of this quickly evolving field.

## Figures and Tables

**Figure 1 cancers-15-00740-f001:**
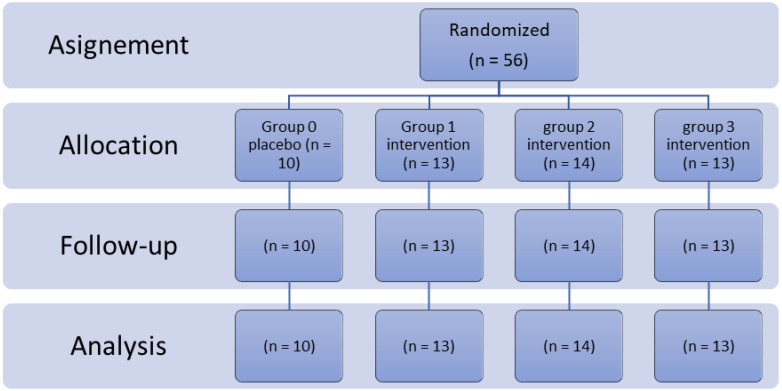
Flow diagram of the study.

**Table 1 cancers-15-00740-t001:** The baseline characteristics of the study population.

	Placebo Group (*n* = 10)	Group 1 (*n* = 13)	Group 2 (*n* = 14)	Group 3 (*n* = 13)	Total (*n* = 50)
Female, *n* (%)	7 (70%)	10 (77%)	13 (93%)	11 (85%)	41 (82%)
Age (years), mean (SD)	47.60 (15.55)	41.08 (10.63)	48.71 (11.24)	39.85 (8.39)	44.20 (11.78)
Weight (kg), mean (SD)	78.80 (31.82)	70.38 (9.01)	80.43 (14.99)	66.54 (5.32)	73.88 (17.52)
Height (cm), mean (SD)	163.60 (9.97)	166.85 (11.10)	159.71 (5.09)	162.46 (8.68)	163.06 (8.99)

**Table 2 cancers-15-00740-t002:** The administered I-131 dose received by patients.

Administered I-131 Activity (mCi)	Number of Patients (%)
100	16 (32.0)
125	4 (8.0)
150	17 (34.0)
175	7 (14.0)
200	6 (12.0)

**Table 3 cancers-15-00740-t003:** Number of treatments for the population.

Number of Treatments	*n* (%)
1	44 (88.0)
2	3 (6.0)
3	1 (2.0)
4	2 (4.0)

**Table 4 cancers-15-00740-t004:** The odds ratio and *p* values of different clinical outcomes for the GEE model. Each intervention group was compared with the placebo.

Outcome Variable	Group 1		Group 2		Group 3	
	Exp (β) (95% CI)	*p* Value	Exp (β) (95% CI)	*p* Value	Exp (β) (95% CI)	*p* Value
pain and swelling in neck	0.852 (0.210, 3.456)	0.823	1.070 (0.276, 4.150)	0.922	0.908 (0.238, 3.457)	0.887
nausea and vomiting	0.763 (0.190, 3.073)	0.704	0.329 (0.080, 1.352)	0.123	0.403 (0.096, 1.695)	0.215
salivary gland swelling	0.943 (0.165, 5.380)	0.948	3.311 (0.384, 28.521)	0.276	2.983 (0.341, 26.103)	0.323
taste loss or change	0.865 (0.178, 4.214)	0.858	0.294 (0.057, 1.514)	0.143	0.591 (0.127, 2.747)	0.502
dry mouth	1.565 (0.362, 6.769)	0.549	0.268 (0.068, 1.061)	0.061	0.781 (0.179, 3.413)	0.743
diarrhea	2.514 (0.523, 12.078)	0.250	2.506 (0.494, 12.713)	0.267	8.665 (0.785, 95.668)	0.078

**Table 5 cancers-15-00740-t005:** The incidence of the side effects in each group in “number of patients”. Each cell contains four lines corresponding to four groups: 0, 1, 2, and 3, respectively. The forward slashes separate the time points in the order of 24/72/120 h after treatment.

	Pain and Swelling in Neck (24/72/120)	Nausea and Vomiting (24/72/120)	Salivary Gland Swelling (24/72/120)	Taste Loss or Change (24/72/120)	Dry Mouth (24/72/120)	Diarrhea (24/72/120)
Strongly disagree	4/4/5	3/4/5	6/8/8	6/7/7	4/4/5	5/6/7
	5/6/7	3/5/7	9/10/10	6/9/10	5/8/8	9/11/11
	6/6/7	1/2/3	12/13/13	4/4/6	2/1/2	11/10/12
	6/5/8	3/3/6	11/12/12	7/8/8	5/5/5	12/12/12
Disagree	2/2/3	3/2/3	3/1/2	2/1/2	2/2/1	1/2/3
	3/0/3	4/3/2	2/1/1	3/1/1	4/2/1	3/0/0
	4/3/5	4/3/9	1/0/0	4/5/7	1/2/5	0/3/2
	1/1/3	0/1/4	1/0/0	1/1/0	1/2/3	0/1/1
Neither agree nor disagree	1/1/1	0/0/0	0/0/0	0/0/0	0/1/0	1/1/0
	0/0/1	0/0/0	0/1/1	1/0/0	0/0/0	0/0/0
	0/0/1	0/1/0	1/1/1	2/1/0	2/1/1	0/1/0
	1/1/1	0/0/0	0/1/1	0/0/0	1/0/0	0/0/0
Agree	3/3/1	4/4/2	1/1/0	2/2/1	4/3/4	3/1/0
	5/7/2	4/3/3	2/1/1	3/3/2	4/3/4	1/2/2
	3/4/1	6/6/2	0/0/0	3/3/0	8/10/6	3/0/0
	4/5/0	8/8/3	1/0/0	5/ 4/5	6/6/5	1/0/0
Strongly agree	0/0/0	0/0/0	0/0/0	0/0/0	0/0/0	0/0/0
	0/0/0	2/2/1	0/0/0	0/0/0	0/0/0	0/0/0
	1/1/0	3/2/0	0/0/0	1/1/1	1/0/0	0/0/0
	1/1/1	2/1/0	0/0/0	0/0/0	0/0/0	0/0/0
Total	10	10	10	10	10	10
	13	13	13	13	13	13
	14	14	14	14	14	14
	13	13	13	13	13	13

## Data Availability

The data presented in this study are available on request from the corresponding author.

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
