# Peer review of "Probiotic Bacteria Cannot Mitigate the Adverse Effects of Radioactive Iodine-131 Treatment"

_cancers, 2023, doi:10.3390/cancers15030740_

Round 1

Reviewer 1 Report

Discussion should be impoved.

RAI theraphy associated diarrhea can be related to TSH levels prior to RAI administration. There is a signifiative difference in your population? 

You affirm that diarrhea should be linked to Iodine 131 activity administered. A biological half life diffence between patients could be involved. Have you quantified activity retention before patients dimission? 

Reviewer 2 Report

Some suggestions to the paper:

1. In the introduction the autors mention that radioiodine treatment can cause constipation as side effect. Please clerify this strange statement.

2. In Figure 1. The placebo group is mentioned as Group 1 later  Group 0. Please harmonise the name of groups consequently in the paper.

3.  The side effects of the treatment are dose dependent. What was the activity of iodine given? Was it a single treatment with fix dose?

4. In the presentation of the results the incidence of the side effects in the groups as absolute number could help to better understand the results.

5. The differences between the groups were not significant but not only dry mouth show changes in group2 but there was some changes in the frequency of diarrhae in group 3 as well. Could you comment it? 

Round 2

Reviewer 2 Report

Corrections OK